# Programming ultrasensitive threshold response through chemomechanical instability

Young-Joo Kim [1], Junho Park[2], Jae Young Lee [2] & Do-Nyun Kim [1,2,3 ✉]

The ultrasensitive threshold response is ubiquitous in biochemical systems. In contrast, achieving ultrasensitivity in synthetic molecular structures in a controllable way is challenging. Here, we propose a chemomechanical approach inspired by Michell's instability to realize it. A sudden reconfiguration of topologically constrained rings results when the torsional stress inside reaches a critical value. We use DNA origami to construct molecular rings and then DNA intercalators to induce torsional stress. Michell's instability is achieved successfully when the critical concentration of intercalators is applied. Both the critical point and sensitivity of this ultrasensitive threshold reconfiguration can be controlled by rationally designing the cross-sectional shape and mechanical properties of DNA rings.

[1] Institute of Advanced Machines and Design, Seoul National University, Seoul, Korea. [2] Department of Mechanical Engineering, Seoul National University, Seoul, Korea. [3] Institute of Engineering Research, Seoul National University, Seoul, Korea. ✉email: dnkim@snu.ac.kr

Constructing an artificial molecular structure with an ultrasensitive threshold response, characterized as a switch-like sigmoidal input-output relationship, has been one of the key challenges in the field of synthetic molecular self-assembly[1–9]. It is ubiquitous in numerous biochemical processes and serves as an essential building block to generate emergent behaviors for cellular regulation[10–13]. Molecular titration is a typical mechanism of ultrasensitivity found in natural molecular networks[14–16]. It occurs when input molecules are sequestered or buffered in inactive complexes by inhibitors, and then an ultra-sensitive threshold response emerges after all inhibitors are consumed. The simplicity of molecular titration renders it attractive for encoding ultrasensitivity into synthetic building blocks. However, controlling the trigger point and the sensitivity of a threshold response independently is limited because they are affected by the concentration and affinity of inhibitors[16,17], but modifying the latter is often challenging.

Here, we propose a chemomechanical mechanism for realizing the ultrasensitive threshold response at the molecular level with highly programmable operating conditions and ranges. We were inspired by Michell's instability[18,19], which indicates that a topologically constrained ring can be suddenly reconfigured when the torsional stress inside the structure reaches a critical value. We construct molecular rings using the DNA origami method[20–22], and torsional stress is applied to them by a DNA intercalator. In this study, we employ ethidium bromide (EtBr), doxorubicin (DOX), and dimeric cyanine dye oxazole yellow (YOYO-1) as representative intercalators. Applying critical concentrations of intercalators successfully initiates Michell's instability at the molecular scale. We are able to control the critical point and the sensitivity of this ultrasensitive threshold reconfiguration by rationally designing the cross-sectional shape and the mechanical properties of DNA rings.

## Results and discussion

**Chemomechanical instability.** We employed a buckling-based structural bifurcation of a ring structure, called Michell's instability (Fig. 1a)[18,19]. The ring structure is formed by joining two ends of a straight rod with a certain amount of twist. Both ends are topologically constrained to prevent their free rotation. The system maintains a planar ring conformation as long as the applied twist ($\theta$) is below the critical value ($\theta_{cr}$). As shown in the result of finite element (FE) analysis for the ring (Fig. 1b and Supplementary Note 1), the applied twist is stored purely as torsional strain energy in the structure, resulting in zero writhe ($W_r$) and a constant in-plane bending strain energy. However, when it exceeds $\theta_{cr}$, the ring structure suddenly becomes super-coiled with a nonzero writhe due to Michell's instability, where the stored torsional strain energy is drastically transformed into the out-of-plane bending energy. Hence, rings are capable of buffering a certain amount of torsional strain energy, enabling a switch-like, threshold reconfiguration from a circle to a supercoil under varied twist angles.

To investigate Michell's instability at the molecular level, we constructed ring structures using the DNA origami method[20–22] (Fig. 1c). Various cross-sections and curvatures can be easily programmed by designing the sequences of short single-stranded DNAs that fold a long, viral single strand into a ring. We carefully designed the structures to minimize unwanted torsional prestress due to the interplay between bending and torsion[21]. DNA intercalators were employed as chemomechanical stimuli to induce torsion in the ring. They are known to perturb the canonical geometry of B-form double-stranded DNA (dsDNA), especially lowering the twist angle between neighboring base pairs (BPs) by 23, 24, and 26 degrees for DOX, YOYO-1, and EtBr,

respectively[23–25]. Since neighboring DNA helices are crosslinked, the geometrical perturbation of dsDNA caused by their binding induces torsional stress in the ring structure, eventually resulting in positive (i.e., left-handed) supercoiling when it reaches its critical value (Fig. 1d). Hence, DNA origami rings exhibit a steep conformational change due to chemomechanically driven Michell's instability. Note that straight DNA origami structures, unlike topologically constrained rings, would be twisted gradually along the helical axis to relieve torsional stress without buckling in response to intercalators[26–28].

**Ultrasensitive threshold reconfiguration.** To demonstrate this, we built six-helix-bundle (6HB) and ten-helix-bundle (10HB) DNA origami rings with and without topological constraints (Fig. 2a). Open ring structures showed wide distributions in the radii of curvature, which corresponded to $90.7 \pm 11.4$ and $51.1 \pm 6.9$ nm for 6HB and 10HB, respectively. Topologically constrained, closed rings were formed by connecting both ends of open rings with additional DNA single strands, resulting in much narrower distributions in the radii of curvature, which were measured as $61.7 \pm 0.9$ and $35.1 \pm 1.4$ nm for 6HB and 10HB, respectively. For all structures, high folding yields of well-folded monomers were achieved (Supplementary Fig. 1).

We first used atomic force microscopy (AFM) images to analyze the conformational changes of these ring structures in response to systematically varied concentrations of EtBr in a model system, since EtBr is a mono-intercalator inducing the highest unwinding to DNA among the three intercalators employed in this study. The binding density of EtBr for DNA nanostructures was ~0–0.2 molecules per BP in the range of concentrations we investigated[27]. We measured the ratio of noncircular (coiled or twisted) structures ($R_{NC}$) among well-folded monomers (Fig. 2b and Supplementary Figs. 2–19). For open rings, $R_{NC}$ gradually increased with EtBr concentration. Because both ends could be freely rotated, they became left-handed helical coils with a gradually increasing twist rate as the concentration of EtBr increased (Fig. 2c). In contrast, ultra-sensitive threshold responses were clearly seen in the $R_{NC}$ curves for closed rings. Almost no change in $R_{NC}$ was observed for EtBr concentrations below the critical concentrations, which were $1 \mu M$ for 6HB and $4 \mu M$ for 10HB, and then, $R_{NC}$ spiked and quick saturation occurred. At these critical concentrations, ~43–58% of monomers were suddenly transformed into super-coiled conformations. The effective Hill coefficient ($N_H$) was estimated by fitting $R_{NC}$ curves to the Hill function to quantify the sensitivity of the threshold response[29,30]. Closed 6HB and 10HB rings showed coefficient values of 11.1 and 10.3, respectively, representing high levels of ultrasensitivity upon EtBr binding[12,31], while open rings exhibited low values of 1.8 and 2.4, respectively. Similar trends were observed using the radius of gyration ($R_g$) of ring structures at various EtBr concentrations (Supplementary Fig. 20).

While 6HB and 10HB closed rings showed similar levels of ultrasensitivity, their critical concentrations for triggering the buckling transition were quite different. This primarily arose from differences in their mechanical properties, particularly in the ratio of bending rigidity ($B$) to torsional rigidity ($C$). A closed ring with a higher $B/C$ ratio would require a higher torsional stress to initiate buckling[18,19]. Generally, the bending rigidity is known to be proportional to $N^2$, where N is the number of helices in the bundle, while the torsional rigidity increases linearly with respect to N[32–34]. Accordingly, the $B/C$ ratio is approximately propor-tional to N, as was also confirmed computationally by normal mode analysis (NMA) (Fig. 2e and Supplementary Note 3). Hence, the critical concentration of EtBr for the 10HB closed ring

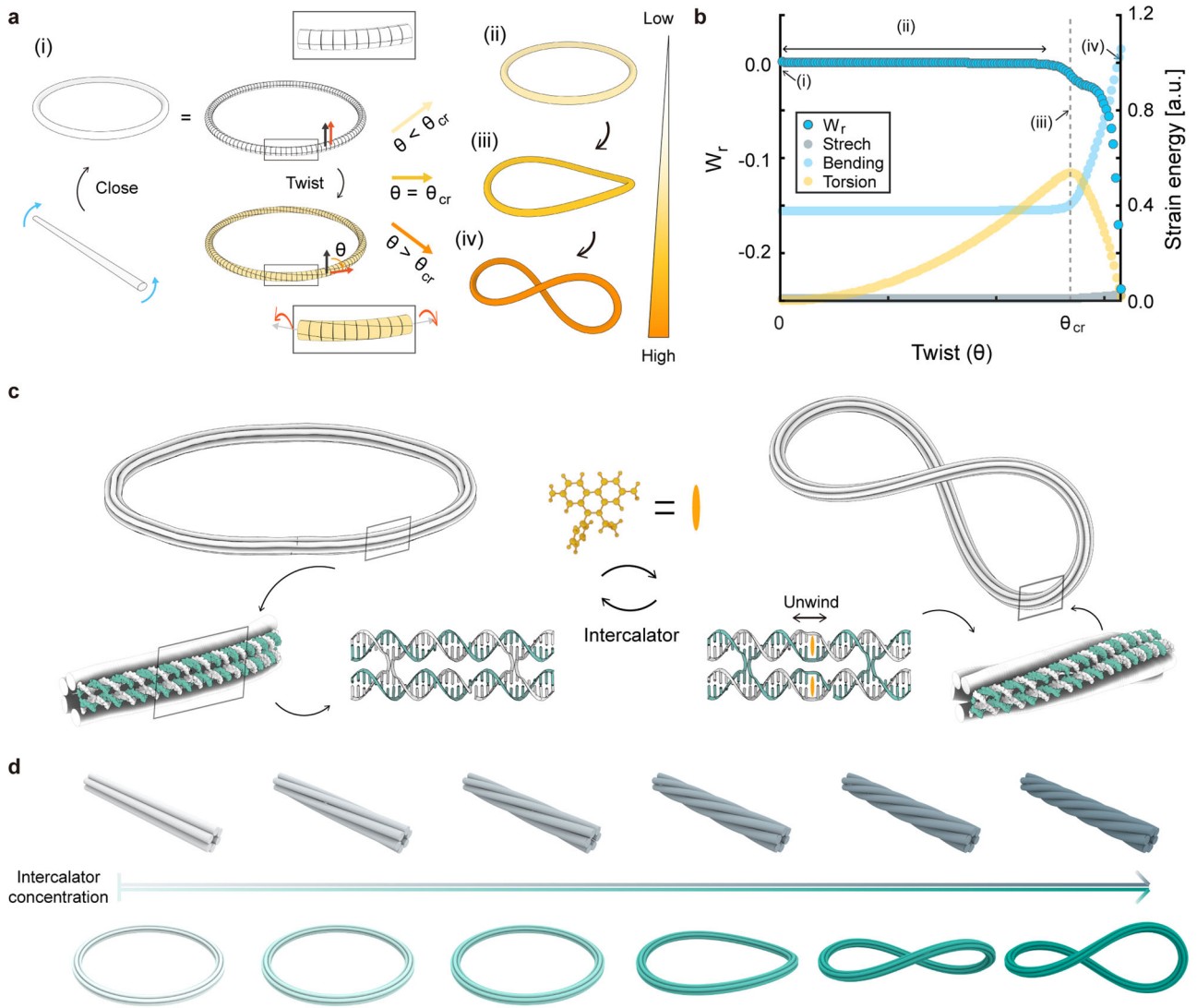

**Fig. 1 Chemomechanical instability of DNA rings. a** Michell's instability. A ring structure is formed by joining both ends of a straight rod with a certain amount of twist. Color bar represents the applied twist angle (θ). $\theta_{cr}$ denotes the critical angle triggering the instability. **b** Writhe ($W_r$) and the strain energy of the ring as a function of the applied twist angle. **c** Realizing Michell's instability with DNA rings. Intercalators perturb the geometry of DNA when bound and induce torsional stress on the ring. **d** A topologically unconstrained straight bundle (top) shows a gradual reconfiguration with respect to the concentration of an intercalator, while an ultrasensitive threshold reconfiguration occurs for a topologically constrained ring structure (bottom).

would be higher than that for the 6HB closed ring. This result shows the great flexibility of the proposed method for controlling the threshold response. We are able to tailor the critical concentration for the buckling transition through the structural design of DNA bundle rings while maintaining the level of ultrasensitivity, which has been well established in structural DNA nanotechnology[21,22].

Moreover, the proposed ultrasensitive reconfiguration was reversible, suggesting that it was mostly based on elastic deformation. $R_{NC}$ of the 6HB ring was measured after adding 2 μM EtBr and removing it by buffer exchange (Fig. 2f and Supplementary Fig. 21)[35]. The results showed that even one buffer exchange was enough to return the supercoiled structure back to its planar conformation, and the reconfiguration was repeatable.

Other intercalators can be employed as chemomechanical stimuli. As each of them has different binding affinities and induces different geometrical perturbations in DNA, the trigger point and the sensitivity of the threshold response to it are expected to be different. We tested DOX, a mono-intercalator

often used as an anticancer drug, and YOYO-1, a bis-intercalator widely used for green fluorescence (Fig. 2g and Supplementary Figs. 22–27). Ultrasensitivity was clearly observed for these agents, with a Hill coefficient of 9.2 for both DOX and YOYO-1. The response curve for DOX was similar to that for EtBr, probably because they have the same intercalation mode (mono-intercalator) and similar binding affinity to DNA[36,37]. On the other hand, YOYO-1 generated the buckling transition at a concentration that was an order of magnitude lower than those of EtBr and DOX. Its distinct intercalation mode (bis-intercalation) and higher binding affinity to DNA[25,36,37] might explain this result, since all three agents effect unwinding of DNA to a similar extent after binding.

**Sensitivity modulation.** The driving force for the reconfiguration of DNA origami rings is chemomechanical (torsional) stress arising from geometrical perturbation in DNA duplexes induced by intercalators. Recently, it was shown that mechanical stress in DNA bundles could be relaxed systematically by introducing short single-stranded regions called gaps into the structure[38].

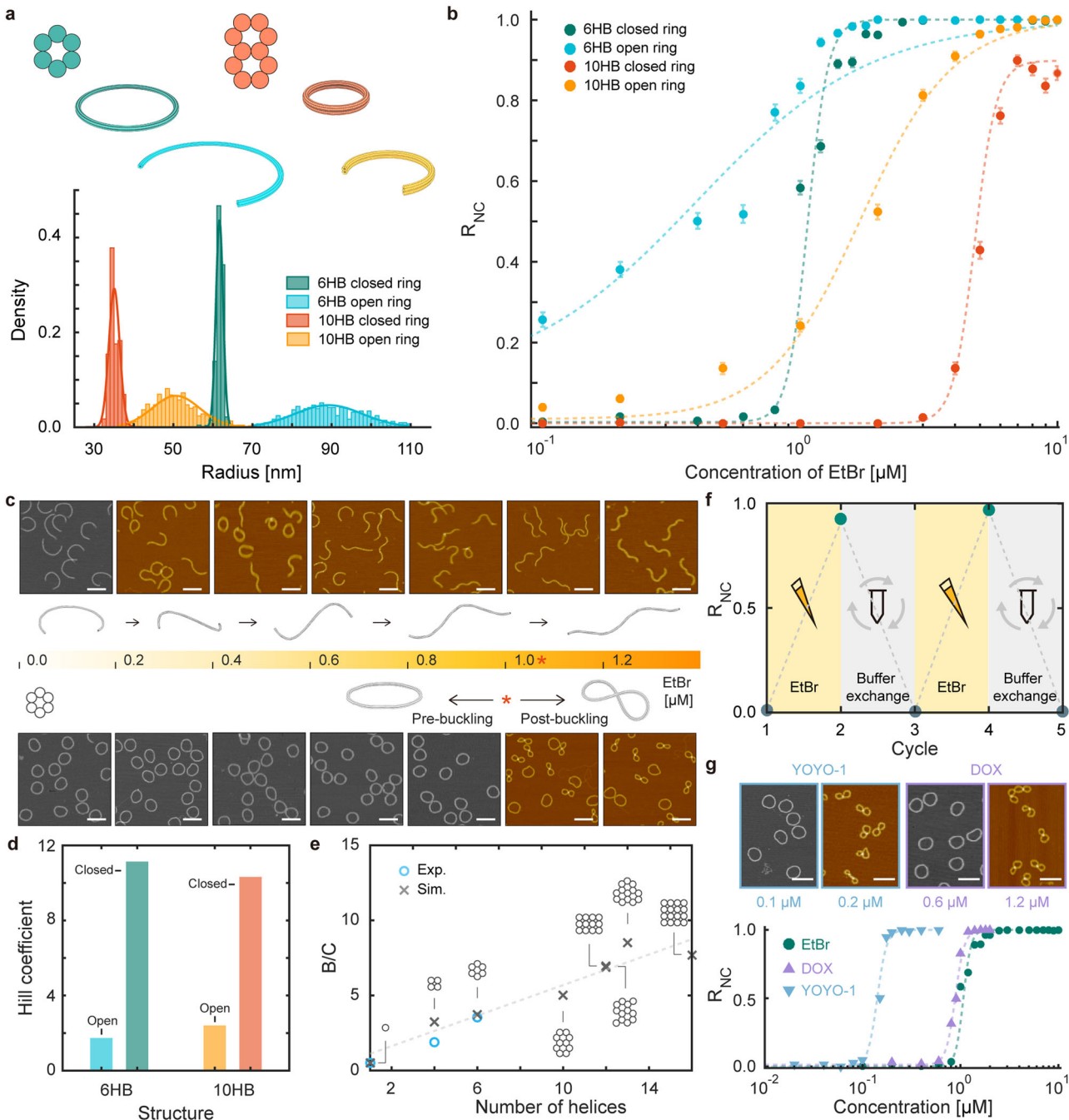

**Fig. 2 Reconfiguration of open and closed DNA rings. a** Schematics and histograms for experimentally measured radii of curvature for 6HB and 10HB rings. **b** Ratio of noncircular monomers ($R_{NC}$) as a function of EtBr concentration. Circles and error bars represent the mean and the standard deviation of experimentally measured $R_{NC}$ values, respectively. Dashed lines represent the fitted Hill curves. **c** Representative AFM images and predicted configurations (Supplementary Note 2) of 6HB open (top) and closed (bottom) rings. Scale bars in the AFM images represent 200 nm. **d** Effective Hill coefficients. **e** Ratio of bending rigidity to torsional rigidity (B/C) of DNA bundles with various cross-sectional shapes. Blue circles and black crosses represent experimental[32,55] and predicted values, respectively. The dashed line shows a linear fit to the predicted B/C values. **f** Reversibility of ultrasensitive reconfiguration. **g** Ultrasensitive threshold reconfiguration of 6HB closed rings by DOX and YOYO-1. Markers and error bars represent the mean and the standard deviation of experimentally measured $R_{NC}$ values, respectively. Dashed lines represent the fitted Hill curves. Scale bars in the AFM images represent 200 nm.

Since the sensitivity of reconfiguration is governed by the level of torsional stress induced by intercalators, we might be able to control it by rationally designing the location and proportion of these gaps in the structure.

To explore this hypothesis, we constructed 6HB rings by placing gaps at nick (single-strand break) positions (Fig. 3a). To minimize undesired structural changes, nicks were placed on helices without any geometrical perturbation and were replaced

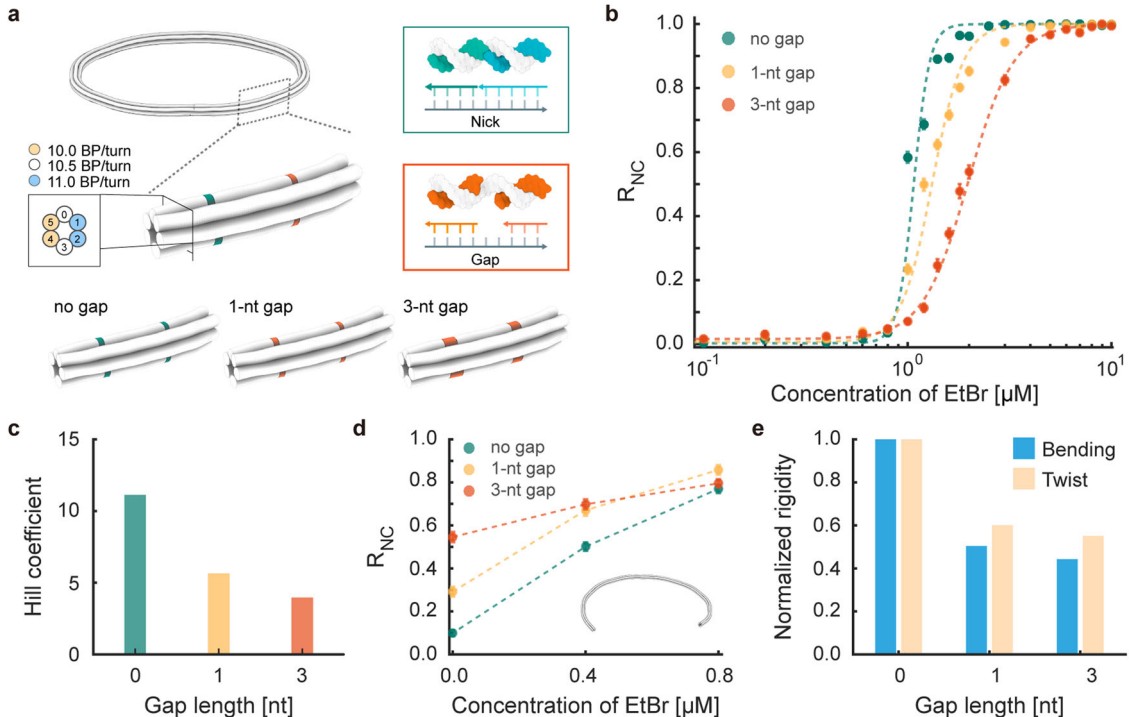

**Fig. 3 Sensitivity modulation through stress relaxation by gaps. a** 6HB closed rings with gaps. **b** $R_{NC}$ values as a function of EtBr concentration. Circles and error bars represent the means and the standard deviations of experimentally measured $R_{NC}$ values, respectively. Dashed lines represent the fitted Hill curves. **c** Effect of the gap length on the effective Hill coefficient. **d** Effect of gaps on the configurational change of 6HB open rings. **e** Effect of gaps on the mechanical properties of straight 6HB structures.

with gaps only when the crossover spacing was 14 BPs long to guarantee sufficient binding energies of staple strands at the gap positions necessary for high folding yield[39,40]. The level of stress relaxation was adjusted by the gap length. We used one-nucleotide (1-nt) and three-nucleotide (3-nt) gaps, which did not deteriorate the structural integrity of 6HB rings (Supplementary Fig. 28). Analyses of AFM images for these structures at various EtBr concentrations revealed that gaps lowered the sensitivity of reconfiguration while maintaining its threshold response (Fig. 3b and Supplementary Figs. 29–38). Hill coefficients were reduced from 11.1 to 5.6 and 3.9 when 1-nt and 3-nt gaps were used, respectively (Fig. 3c). As we can vary the length, location, and number of gaps broadly[38,40], the level of stress relaxation and hence the sensitivity of reconfiguration can be controlled more widely and finely.

Note that the critical concentration of EtBr triggering the buckling transition was not changed much by gaps, unlike the sensitivity of reconfiguration. This was counterintuitive because stress relaxation by gaps would require a higher concentration of EtBr to induce the torsional stress required for initiating Michell's instability. Hence, there must be other effects of gaps on the structure that nullify the effect of stress relaxation on the critical point. We found two potential explanations for the minimal change in the critical concentration of EtBr caused by gaps. First, we investigated the conformational changes of 6HB open rings with and without gaps with respect to EtBr concentration. Unlike closed rings with topological constraints, large portions of open rings with gaps (30 to 55%) were folded into nonplanar, helical coils with left-handedness even without the presence of EtBr (Fig. 3d and Supplementary Figs. 39 and 40). The portion of helical coils was higher when gaps were used and monotonically increased with increasing EtBr concentration. This suggests that the left-handed twist was more readily induced by gaps in the open ring and gaps created torsional prestress when topologically

constrained into closed planar rings; this lowered the chemo-mechanically applied critical twist required for the buckling transition. Second, gaps affect the mechanical properties of DNA bundles. Normal mode analysis using FE models for straight 6HBs with and without gaps revealed that both torsional rigidity (C) and bending rigidity (B) were decreased by gaps (Fig. 3e and Supplementary Note 3)[40]. In particular, the B/C ratio decreased with gaps as the reduction rate of B was greater than that of C. As a lower B/C ratio requires a lower torsional stress for Michell's instability, it lowers the required critical concentration of EtBr[18,19]. These two structural effects caused by gaps (generation of torsional prestress and reduction of the B/C ratio) might compensate for the effect of stress relaxation, resulting in barely noticeable differences in the critical concentration. In fact, this offers a versatile way of controlling the threshold response via structural design. We may independently modulate the critical concentration of chemomechanical stimuli by using cross-sectional design of rings and the sensitivity by gaps or engineered defects[40].

**Importance of shape homogeneity.** When we design an ultra-sensitive threshold response based on Michell's instability, both the topological constraint forming a closed structure and the homogeneity of the structural shape in the ring are important. To illustrate, we designed a 6HB closed triangle using the same cross-sectional shape and the bundle length of the 6HB closed ring (Supplementary Fig. 41). The triangle has straight edges and curved vertices, and hence, the deformation energy is localized near the vertices in contrast to the ring, where it is rather homogeneously distributed throughout the structure. The 6HB closed triangle showed a gradual reconfiguration upon EtBr binding similar to those of open rings, but without a threshold (Fig. 4a and Supplementary Figs. 42–56). Local coiling modes

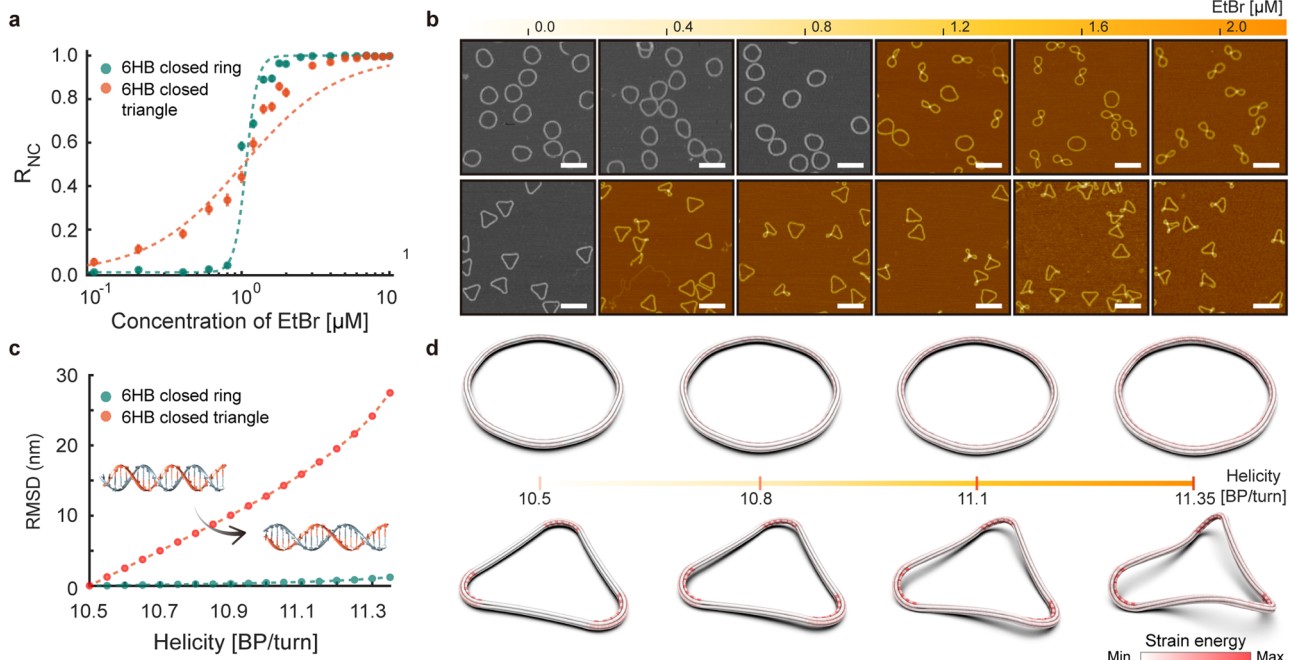

**Fig. 4 Differences in the reconfiguration responses between rings and triangles. a** $R_{NC}$ values as a function of EtBr concentration. Circles and error bars represent the means and the standard deviations of experimentally measured $R_{NC}$ values, respectively. Dashed lines represent the fitted Hill curves. **b** AFM images of 6HB closed rings (top) and triangles (bottom). Scale bars in the AFM images represent 200 nm. **c** Root-mean-squared displacement (RMSD) of 6HB closed rings (green) and triangles (red) predicted by FE analysis. RMSD was calculated for structures with perturbed helicities with respect to the reference structure exhibiting a helicity of 10.5 BP/turn. **d** Simulated conformations of 6HB closed rings (top) and triangles (bottom). The color bar represents the strain energy stored in BPs.

emerged at the vertices even at low concentrations of EtBr without any buffering of induced torsional strain energy (Fig. 4b).

FE analysis for closed ring and triangle structures captured these behaviors as well. Their three-dimensional shapes and strain energies were calculated by varying the helicity of dsDNA to consider the effect of EtBr binding (Supplementary Note 2). The root-mean-square-displacement (RMSD) curves showed a remarkable difference between the two structures in terms of conformational changes with respect to helicity (Fig. 4c). The closed ring maintained its circular shape in a plane with no significant conformational change, and its strain energy almost uniformly increased as the torsional energy throughout the structure increased (Fig. 4d). In contrast, the closed triangle was gradually reconfigured by changes in helicity, with local coiling occurring near its vertices where much higher strain energy was concentrated. The triangular structure could not maintain a planar configuration with distinct deformation energies induced at vertices and edges while still satisfying geometrical compatibility under the topological constraint. Therefore, it would be essential to design a structure with shape homogeneity so that chemomechanically induced deformation energies could be distributed uniformly throughout the structure to achieve an ultrasensitive threshold response.

We showed that ultrasensitive threshold responses could be programmed into synthetic molecular structures *via* chemomechanical instability without inhibitor molecules. Due to Michell's instability, topologically constrained molecular rings were capable of buffering chemomechanically induced stresses without conformational change. While Michell's instability was used to explain the supercoiling of dsDNA rings theoretically[41], herein, we apply it to structured DNA assemblies in a highly programmable manner. The trigger point and the sensitivity of reconfiguration could be easily modulated by controlling the

global and local mechanical properties of rings for a given intercalator.

The proposed mechanism might be realized with other DNA structuring methods, such as DNA tile[42] and brick methods[43], and it could also be applicable to other engineered molecular self-assemblies of RNA[44,45] or proteins[46,47]. While we used intercalators as a chemomechanical stimuli in this study, any other DNA binding molecules that perturb the geometry of dsDNA, including DAPI, topotecan, netropsin, and cisplatin, can be used alternatively to induce chemomechanical instability[27,37,48]. The use of multiple chemical agents with different binding affinities and structural effects on DNA would broaden the tunable range of ultrasensitive threshold responses even further. Environmental factors, which can affect the binding affinity of chemomechanical stimulus agents, such as salt concentration or pH, can be adopted as additional tuning methods[49].

Our reconfiguration mechanism would be further utilized to realize ultrasensitivity in other chemical or physical responses by precisely arranging functional or interacting molecules onto DNA nanostructures and using the proximity-induced reactions between them after reconfiguration[50–52]. In addition, DNA rings might be embedded in soft materials such as DNA hydrogels, where they could serve as crosslinkers for scaffold polymers to enable ultrasensitive threshold deformations cause by stimuli.

## Methods
**Self-assembly of DNA origami structures**. Using an open source program, caDNAno[53], all structures used here were designed on the honeycomb lattice with a M13mp18 scaffold strand (7249-nt-long, GUILD, www.guildbioscience.com). Sequences of staple strands for the structures were exported from the caDNAno (Supplementary Tables 1–5) and they were synthesized from Bioneer (www.bioneer.co.kr). A folding mixture consists of 20 nM concentration of scaffold DNA, 100 nM concentration of each staple strand, 1 × TAE buffer (40 mM Tris-acetate and 1 mM EDTA, Bioneer) and 20 mM of MgCl₂ (Sigma-Aldrich,

www.sigmaaldrich.com). The annealing process for self-assembly of DNA strands was performed by establishing temperature gradients from 80 to 60 °C with a rate of −0.25 °C/min and from 60 to 45 °C at a rate of −1 °C/hr in a thermocycler (T100, Bio-Rad, www.bio-rad.com). Excessive staple strands were removed through five buffer exchange procedures[35] at 5 krcf during 8 min and concentration of structures was adjusted using the same buffer used in folding (1 × TAE and 20 mM of MgCl$_2$). Concentrations of folded structures were measured using a Nanodrop One UV spectrophotometer (Thermo Fisher Scientific, www.thermofisher.com). Purified structures were stored at −4 °C in a refrigerator.

**Agarose gel electrophoresis.** Annealed samples of DNA origami structures were electrophoresed using 1% agarose gel containing 0.5x TBE (45 mM Tris-borate and 1 mM EDTA, Sigma-Aldrich), 12 mM MgCl$_2$, and 0.5 µl/ml EtBr (Noble Bioscience Inc.). Electrophoresis was performed for 120 min at 60 V bias voltage in an ice-water cooled chamber. Gel imaging was performed using GelDoc XR + device and Image Lab v5.1 program (Bio-Rad).

**AFM imaging.** Before deposition, the purified sample was diluted by the folding buffer solution and mixed with varying amount of intercalators. 0.4 nM annealed structures were used in order to ensure appropriate numbers of monomers on the substrate for image analysis. The 20 µl of diluted sample was then deposited and incubated on a freshly cleaved mica substrate (highest grade V1 AFM Mica, Ted-Pella Inc.) for 5–10 min. The substrate was washed with DI water and gently dried using a N$_2$ gun (<0.1 kgf/cm$^2$). If the number of monomers in images was small, the sample was incubated for a longer time. AFM images were taken by a NX10 system (Park Systems, www.parksystems.co.kr) using the noncontact mode in SmartScan software. A PPP-NCHR probe with spring constant of 42 N/m was used in the measurements (Nanosensors). Each image had a sample area of 5 µm × 5 µm in 1024 × 1024 pixel resolution. Generally, five to eight AFM images were obtained for one sample for each case to obtain statistically sufficient numbers of monomer particles (at least 300 particles). Images were flattened with linear and quadratic order using the XEI 4.1.0 program (Park Systems). After removing aggregates through filtering by the pixel size, monomeric particle images of DNA origami structures were extracted from AFM images using custom scripts developed in MATLAB R2016a (MathWorks Inc.) and provided in the literature[54], and classification of single particles was done manually. Only well-folded monomers were used in analyses. Conformations of monomers were manually classified depending on whether they were circular or not. After binarization of collected monomer particles, pixels were fitted to a circle in order to calculate the radius of curvature using custom scripts developed in MATLAB R2016a.

**Effective Hill coefficient.** Sensitivity of reconfiguration of DNA origami nanostructures was analyzed by fitting the Hill equation, expressed as

$$R_{NC} = R_{NC,min} + (R_{NC,max} - R_{NC,min})\frac{x^{n_H}}{x_{50}^{n_H} + x^{n_H}} \tag{1}$$

where x is the concentration of intercalator[30]. Here, $R_{NC,min}$ and $R_{NC,max}$ are the minimum and the maximum $R_{NC}$, respectively, while $x_{50}$ refers to the concentration of intercalator required to reach half-maximal $R_{NC}$. The equation was fitted by minimization of the sum of squared relative residuals using the function 'fmincon' in MATLAB R2016a.

## Data availability

Data supporting the findings of this study are available in the main manuscript and the supporting information. AFM images not included in the supporting information but used in analyses are available from the authors upon reasonable request.

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

## Acknowledgements

This work was supported by the National Convergence Research of Scientific Challenges (NRF-2020M3F7A1094299) and the Basic Research Program (NRF-2019R1A2C4069541) through the National Research Foundation of Korea Foundation of Korea (NRF) funded by Ministry of Science and ICT.

## Author contributions

Y.-J.K. and D.-N.K. conceived the design approach and modeling. Y.-J.K. and J.P. performed the experiments and analyzed the data. J.Y.L. performed FE simulations and Y.-J.K. and J.Y.L. analyzed the data. Y.-J.K. and D.-N.K. discussed the results and wrote the manuscript. All authors commented on and edited the manuscript.

## Competing interests

The authors declare no competing interests.
