## [Peer Review File · Nature Communications]

Programming ultrasensitive threshold response through chemo-mechanical instabilityEditorial Note: This manuscript has been previously reviewed at another journal that is not operating a transparent peer review scheme. This document only contains reviewer comments and rebuttal letters for versions considered at *Nature Communications*.

REVIEWERS' COMMENTS

Reviewer #1 (Remarks to the Author):

The authors have responded to the previously raised comments in a proper and complete fashion. I am extremely pleased to see they included the Yoyo and Dox study and the results are integrated in the main document. To me, this makes the demonstration of versatility much stronger and the story more complete.

Detailed response to the reviewers' comments

Manuscript ID: NCOMMS-21-11115-B

Title: Programming ultrasensitive threshold response through chemo-mechanical instability

Authors: Young-Joo Kim, Junho Park, Jae Young Lee and Do-Nyun Kim

We appreciate useful comments and very positive assessment of our revised manuscript by the reviewers.

Reviewer #1's comments to the authors:

The authors have responded to the previously raised comments in a proper and complete fashion. I am extremely pleased to see they included the Yoyo and Dox study and the results are integrated in the main document. To me, this makes the demonstration of versatility much stronger and the story more complete.

Response to the Reviewer #1's comments:

We would like to express our gratitude to the Reviewer. Thanks to many helpful comments during the review process, we are truly grateful that our work has been improved a lot to better communicate the importance of it.